# Features of Regulation Document Translation into a Machine-Readable Format within the Verification of Building Information Models

Elena Makisha

Department of Information Systems, Technologies and Automation in Construction, Moscow State University of Civil Engineering, 26, Yaroslaskoye Highway, 129337 Moscow, Russia; makishaev@mgsu.ru

**Abstract:** The transition to a design based on information modeling has paved the way for automated verification of project documentation. The most complicated type of design documentation check is the assessment of compliance with the requirements of regulatory documents since its automation requires the translation of statements written in natural language into a machine-readable format. At the same time, building codes and regulations should be suitable for translation into machine-readable form. However, a large number of provisions presented in regulatory documents cannot be subjected to automated verification due to their specific features. This study aimed to analyze the suitability of the regulatory provisions to be translated into a machine-readable format, identify limiting factors, and establish recommendations to avoid these factors. This study was conducted based on the example of the code of rules for "Residential Apartment Buildings" (SP 54.13330.2016) applied in the Russian Federation. During the research, a previously developed algorithm that generates rules for checking building information models and is based on the RuleML language was applied to the provisions of the standard above to identify statements that can and cannot be translated. As a result, 356 statements were received, which were divided into five groups: requirements suitable for translation into a machine-readable format, requirements containing links to other documents (regulatory and not only), requirements of uncertain interpretation, requirements that cannot be verified based on an information model, and informative requirements. For the first group of statements, there were examples given for both the statements and the rules derived from them. For the other four groups, examples of statements were supplied with factors preventing the translation of requirements into a machine-readable format and solutions on how to avoid these factors. An analysis of the distribution of statements (related to the above-mentioned groups) by sections of the standard showed that a significant part of the requirements is suitable for translation into a machine-readable format. The possible obstacles to translation can be overcome by attracting experts and using programming tools. The paper also makes recommendations on how to arrange new standard structures.

**Keywords:** BIM; design assessment; building codes; design guides; rule checking; logic rule

## 1. Introduction

The rapid implementation of information modeling has significantly altered the design process for construction projects. The trend toward greater automation of project procedures has grown stronger, owing to the use of ready-made software solutions, as well as the development of plug-ins or the use of embedded programming tools. In particular, special software tools could verify the design results in the form of building information models.

The operation of verification systems is quite complex and may include the processing of various types of information that are not always suitable. For instance, one of the types of information that is processed during inspections is the regulatory documents and standards written in natural language. This article focuses on regulatory document processing, the results of which can be used to automate the verification of building information models.

*1.1. Types of Building Information Model Checks at the Design Stage*

Draft solutions toward this approach were invented in the 1980s in the shape of expert systems, which were later transformed into distributed knowledge bases [1–4]. The IFC (Industry Foundation Classes) standard, which appeared in the 1990s, became a unified machine-readable description of the projected construction object and made a significant contribution to the development of automated verification systems. Consequently, the verified data about the object could not only be submitted into the system by the user but also automatically extracted from the object description in IFC format [5–8] Various software for checking information models in the IFC format, such as Autodesk Navisworks, Solibri, EXPRESS Data Manager (EDM), BIM Vision, simpleBIM, BIM Model Checker, AllCheck, Rusbimexpert, and others, have appeared since the 2000s [9–16]. In some countries, systems for checking information models for compliance with regulatory requirements and issuing construction permits have been developed independently or on the basis of the above-mentioned tools, including Corenet (Singapore), DesignCheck (Australia), ByggSøk (Norway), SMARTcodes (USA), solutions from Statsbygg (Norway), Design Assessment Tool (USA), SEUMTER (Korea), and ACABIM (New Zealand) [17–35].

Among the existing systems, the main directions for checking information models of construction objects have gradually formed (Figure 1) [36–39].

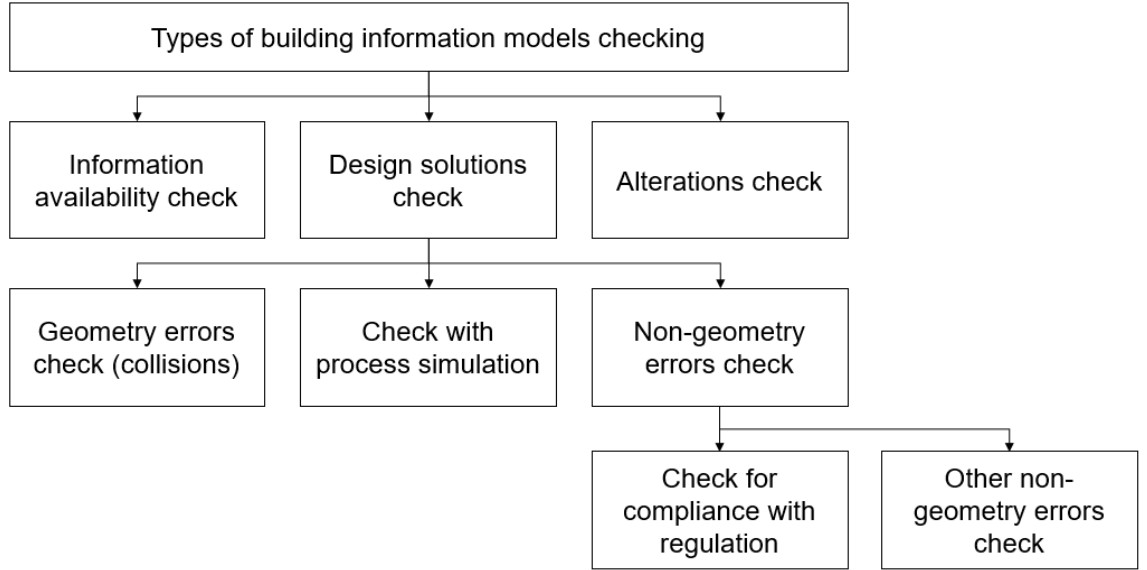

**Figure 1.** Types of automated verification for building information models at the design stage.

As part of the information availability check, the presence of the obligatory elements of the model (availability, completeness, and correctness of property values) is analyzed. Depending on the stage of the life cycle, there may be different requirements for the availability of information in the model.

A modification check involves:

- Comparing the old and new versions of the model;
- Identifying new and deleted elements;
- Comparing the properties of the changed elements.

Checks can be performed by simulation or imitation of a physical process. This type of check can include the calculation of a constructive model, a collision check during 4D planning (modeling of a construction organization project), and a project for the implementation of construction operations.

Checks for compliance with regulatory documents can be different; however, before proceeding to them, basic checks (including those mentioned above), which may be created utilizing program templates or a programming environment, should be passed.

*1.2. Verification for Compliance with Regulatory Documents*

Automated assessment of compliance with the requirements of regulatory documents is one of the most difficult types of checks since it requires translation into a machine-readable format for provisions written in natural language. The provisions of any standard, translated into a machine-readable format, are commonly called rules. To date, several studies have been conducted aimed at creating various systems to verify the requirements of regulatory documents. As a result, three main methods of obtaining rules can be distinguished:

1. Coding using high-level programming languages.

Direct coding of requirements in a programming language is the easiest way to translate the requirements of standards into a machine-readable format. This method is used in Corenet systems (Singapore) [17–19] and EDM (Norway) [11], which in turn was the basis of the DesignCheck program (Australia) [20,21].

The main disadvantage of this rule-obtaining method is the necessity to hire a qualified programming specialist, who may not have sufficient knowledge of the subject area, which can lead to errors in rule coding. In addition, this method requires long-term support by the programmer for adjustments, updates, or expansions of the database of encoded requirements. Another disadvantage is the dependence of encoded requirements on specific proprietary software. In addition, there is a missing link in the source text of the regulatory document for proper updating of encoded requirements.

2. Creating rules based on software templates.

Templates allow users to create new rules by changing the types of elements under check and their parameters without requiring computer programming. This approach is implemented in the Solibri system, which was used, for instance, for automated verification of projects in the US General Services Administration [22] and during the implementation of the HITOS project by the Statsbygg State Administration for Civil Engineering and Real Estate in Norway [23]. Later Solibri developers added the opportunity to program rules in Java through the appropriate API; thus, both approaches can be implemented simultaneously.

This method depends on the rules of specific proprietary software and the missing connection with the source text of the regulatory document for timely updating of encoded requirements. In the case of systems that do not provide access to the API for programming rules, a developer company should be involved to adjust, update, and expand the database of encoded requirements.

3. Implementation of a rules-based language of knowledge representation.

This approach is considered the most promising for applying the rules due to its simplicity and visibility for experts, its high modularity and flexibility for additions and changes, and the transparency of the logical inference mechanism. Methods based on knowledge representation languages have already been implemented in the following systems: SMARTcodes (USA) [32], SEUMTER (Korea) [24–26], and ACABIM (New Zealand) [27–31]. This approach was revealed in several studies on a rules algorithm for verification of building information models that were based on the RuleML language [23,24].

Despite the advantages of the latter method, it, like the previous two, it implies a large amount of manual work. This forced the researchers to invent new approaches for automating the process of creating machine-readable rules.

Fuchs and Amor have reviewed natural language processing tools, which were considered in the context of the tasks they perform [40]. Zhang and El-Gohary have proposed a method for the fully automated extraction of semantic and syntactic elements of information from regulatory documents based on deep neural networks [41]. In the study by Wu et al., a hybrid model of deep learning is proposed to translate into a machine-readable format the limitations that arise when using the AWP (Advanced Work Packaging) methodology [42]. Xue and Zhang developed a new tagger for marking parts of speech (Part-of-Speech-Tagger,

POS-Tagger) adaptable to building codes [43]. It uses a deep learning neural network model and error-based transformation rules.

Nevertheless, the use of neural networks for the automated translation of requirements into a machine-readable format is possible only if the initial training data are available. Training can be conducted based on a certain amount of already marked-up regulatory documents. Obtaining this volume is possible by manually extracting semantic and syntactic elements of information from regulatory documents. Now, there is a lack of such data for training neural networks; in many countries, they are completely absent. The Russian Federation is not an exception since there is no explicit decision yet regarding the use of a particular markup language. In addition, regardless of the method of processing natural text (manual or automated), construction codes and regulations adopted in a particular country should be suitable for translation into machine-readable form. However, many of the provisions presented in regulatory documents have features that prevent this.

The purpose of this study was to develop a text labeling algorithm based on the rule modeling language to obtain rule bases. Its use will be possible both directly for conducting checks and for training neural networks for further automated text markup.

To achieve this goal, the following tasks will be solved:

1.  The choice of the rule modeling language as a tool of the algorithm;
2.  Formation of the methodology for the approval of the algorithm;
3.  Development of an algorithm for forming rules for checking building information models based on the selected rule modeling language;
4.  Automation of the algorithm for forming rules for checking building information models;
5.  Approval of the algorithm for forming rules for checking building information models based on the rule modeling language;
6.  Analysis of the suitability of the developed algorithm for translation into a machine-readable format for regulatory documents;
7.  Analysis of the suitability of the regulatory documents' provisions for translation into a machine-readable format, as well as addressing troubles and recommendations for troubleshooting.

## 2. Materials and Methods

*2.1. Justification of the Choice of the Rule Modeling Language as a Tool of the Algorithm Being Developed*

Ref. [44] revealed prospects for the application of rule-based knowledge representation languages for rule coding and creating rules based on program templates in high-level programming languages. Earlier, this was largely because of the drawbacks of other methods. Now, it seems useful, as the marked-up texts can be applied for further training of neural networks.

The experience with SMARTcodes has been the basis for a number of studies based on the RASE [45] tagging mechanism and its extension based on the DROOLS rule mechanism [46]. Other specialists tried to apply the LINQ query programming language to labeling [47] and used the semantic web techniques OWL and SWRL in combination with the JESS rules engine [48]. However, none of these initiatives received international distribution, mainly due to the lack of an opportunity to establish a relationship between the rule base and the source text of the normative document.

Among the rule-based knowledge representation languages, SMARTcodes, KBim Logic (SEUMTER system), and LegalRuleML (ACABIM system) have already been used. However, SMARTcodes and KBim Logic are local developments adapted to the features of certain standardization systems. They are not openly published standards. The Legal-RuleML language is an open standard related to RuleML. However, LegalRuleML is majorly focused on the legal side and not on the logical side of regulatory documents, which may not be very convenient for further use of the marked-up text for training neural networks [49,50].

Among the rule-based languages, RuleML (developed and maintained by the non-profit organization RuleML Inc., Fredericton, Canada) may be considered an actual standard. It is a system of rule modeling language families used for unified representation and exchange of basic types of rules between different logics and platforms [51–55].

A detailed justification of the RuleML application to translate regulatory documents into machine-readable rules was given in the study by [25]. In the context of the current research, the following advantages should be highlighted:

1. RuleML is based on higher-order logic, which allows for obtaining a formalized representation that is understandable both for the program and for the human;
2. In its syntax, RuleML allows for establishing a connection of rules with the original document's text;
3. Software environments (e.g., Lime or Rawe) can be applied to RuleML;
4. RuleML is the actual rule exchange standard developed and maintained by the non-profit organization RuleML Inc.; its specification is open and published on the official website;
5. The implementation of RuleML is scalable; there are extensions, such as MathML for mathematical rules and LegalRuleML for legal documents;
6. Since RuleML was developed as an exchange format, the rules written for it do not depend on a specific software or hardware platform; it has a long support cycle and can interact with other rule bases;
7. Based on the provisions of first- and higher-order logic, RuleML rules can show an unlimited range of requirements, including multi-level conditions within a certain area of knowledge.

### 2.2. Methodology of Approbation of the Developed Algorithm

A previously developed algorithm for forming rules of verification of building information models based on the RuleML language was used to assess the suitability of the provisions of regulatory documents for translation into a machine-readable format [23,24]. The application of the specified algorithm to the provisions of the standard allowed the identification of the requirements, which could or could not be translated. The limiting factors for those requirements that cannot be presented in machine-readable form were analyzed.

The study was conducted based on the code of rules for "Residential Apartment Buildings" (SP 54.13330.2016), which is applicable in the Russian Federation [56]. The scope of this code is the design and construction of newly constructed and reconstructed multi-apartment residential buildings up to 75 m in height, including dormitories of the apartment type as well as residential premises that are part of buildings for other functional purposes. Moreover, this standard includes various groups of requirements that affect essential aspects of building design, so the investigation of the standard may have a vast practical interest.

The results obtained based on this standard can be extended to other Russian and foreign standards.

### 3. Results

### 3.1. An Algorithm for Forming Rules for Checking Information Models of Construction Objects Based on the Ruleml Rule Modeling Language

The rule-forming algorithm for verification of the building information models based on the RuleML consists of two main stages, as shown in Figure 2. The description of the algorithm is performed in accordance with the EPC notation. An electronic copy of the regulatory document is received at the input of the algorithm, and a set of rules is formed at the output, which is placed in the database.

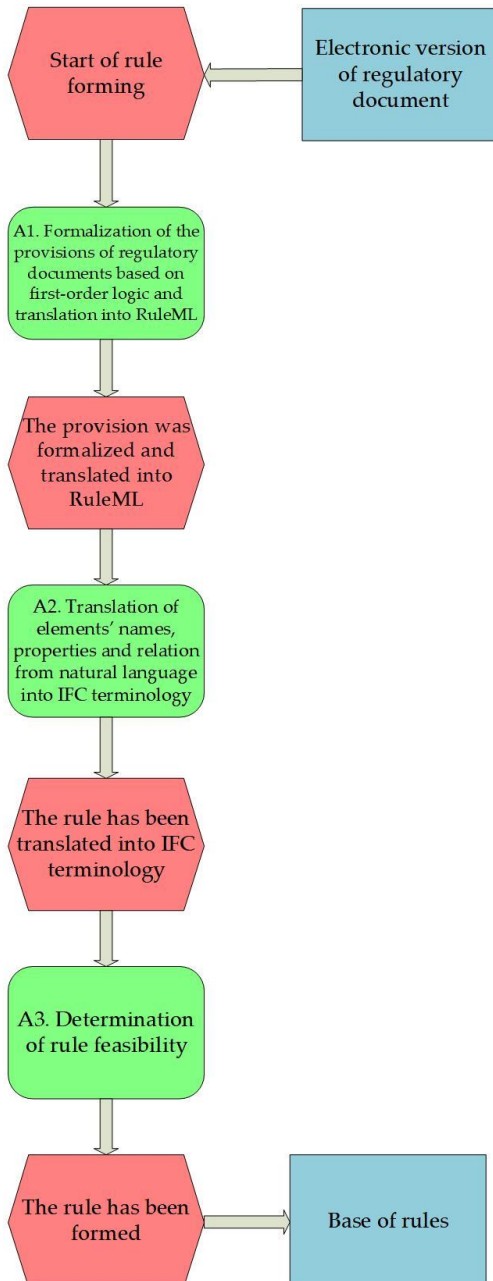

**Figure 2.** The general algorithm of verification of building information models.

A1. At stage A1, the provisions of regulatory documents are formalized based on the logic of the first and higher orders, and the further assignment of the corresponding tags to the identified logical units according to RuleML syntax.

Formalization means to represent the meaningful field (reasoning, evidence, classification procedures, information retrieval, scientific theories) as a formal system or calculus.

Single statements, logical constants, atoms, predicates, and arguments of this statement are consistently identified within a provision (stages A1.1–A1.4, Figure 3). Normally, a provision may contain several statements, i.e., establish several interrelated requirements. Usually (but not always) one sentence of a paragraph corresponds to one statement. Statements are divided into requirements and informative statements.

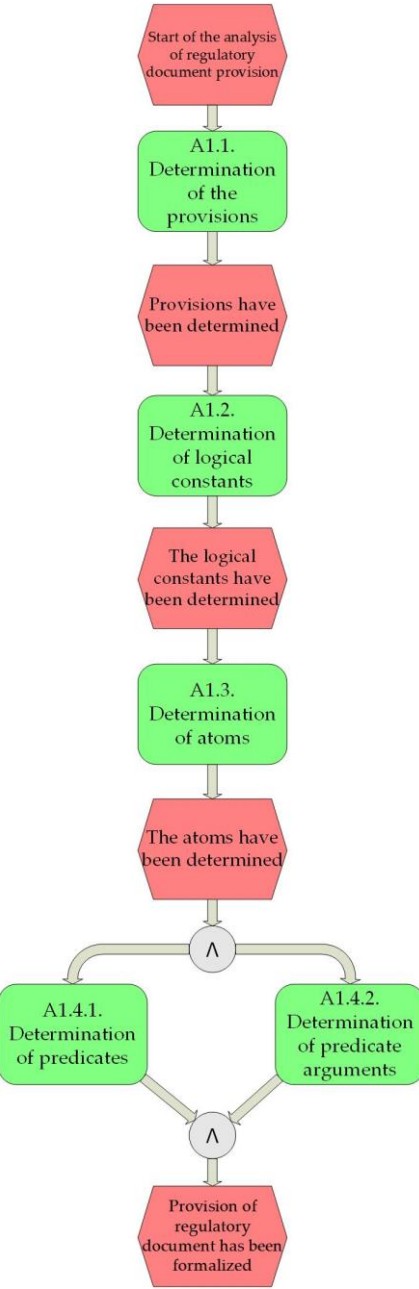

**Figure 3.** The first stage of the general algorithm of verification of building information models.

Logical elements are placed in the appropriate tags provided by RuleML language (Figure 4).

A2. At stage A2, the content of tags is translated into the terminology of the Industry Foundation Classes (IFC) standard [57] to ensure the relationship between the rules and entities, properties, and relationships of the information model (Figure 5).

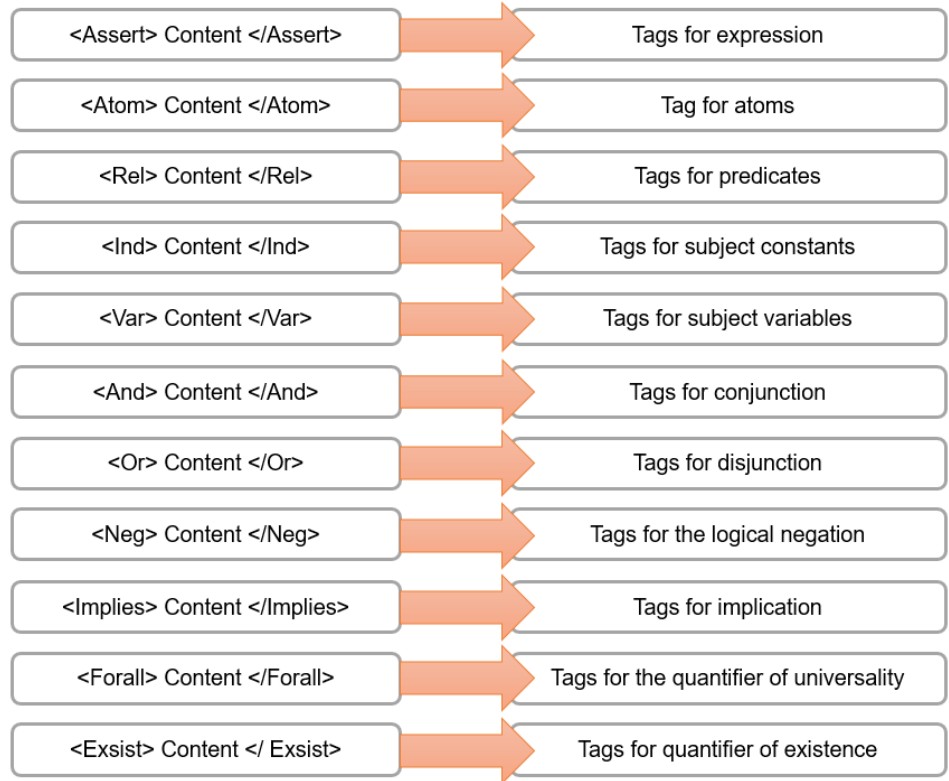

**Figure 4.** Correspondence between RuleML elements and logic elements.

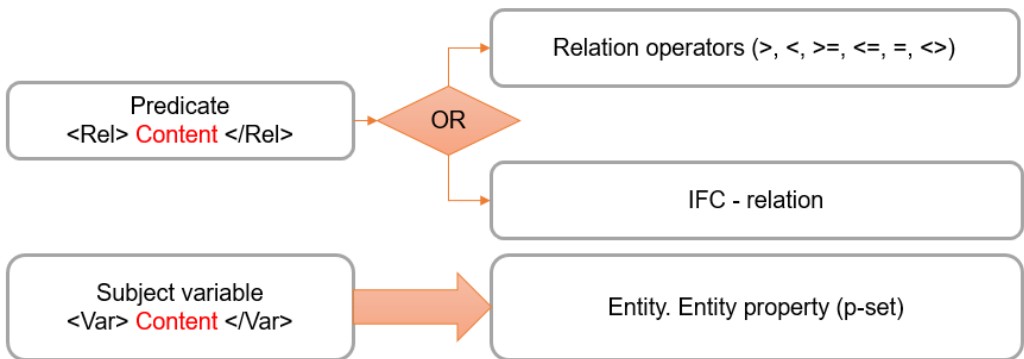

**Figure 5.** Correspondence between tag content and IFC scheme elements.

### 3.2. Automation of the Rule Formation Algorithm for Ruleml-Based Verification of Building Information Models

The RuleML application was supported with both text editors and available development tools, which can partially automate this process. However, there was a specific environment in Excel using Visual Basic for Applications developed with a user-friendly interface. The interface of the developed software module is shown in Figure 6. This tool was developed in Excel using Visual Basic for Applications.

### 3.3. Implementation of the Rules Formation Algorithm for Building Information Model Checks Based on the Rule Modeling Language

Within the study, the code of rules for "Residential apartment buildings" (SP 54.13330.2016), which acts in the Russian Federation, was investigated. The structure of this regulatory document is shown in Table 1.

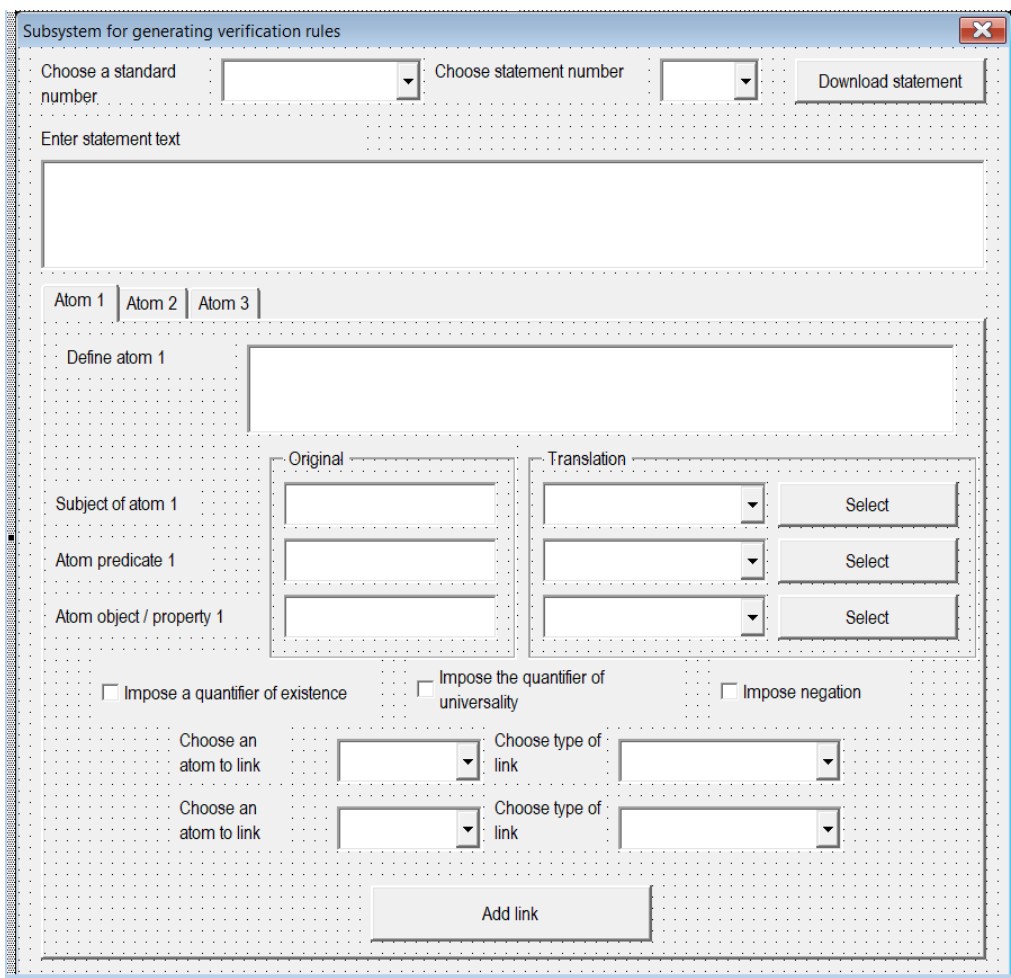

**Figure 6.** The interface of the software module for generating verification rules.

**Table 1.** The structure of the code of rules SP 54.13330.2016.

| Section Number | Section Title | Amount of Provisions in the Section | Amount of Statements in the Section |
|:---:|:---:|:---:|:---:|
| 1 | Aim and scope | - | - |
| 2 | Regulation references | - | - |
| 3 | Terms and definitions | - | - |
| 4 | General terms | 18 | 76 |
| 5 | Requirements for buildings and premises | 9 | 22 |
| 6 | Bearing capacity and permissible deformability | 8 | 16 |
| 7 | Fire safety | 51 | 110 |
| 7.1 | Prevention of fire spread | 16 | 40 |
| 7.2 | Evacuation | 15 | 29 |
| 7.3 | Fire protection requirements for engineering Systems and equipment of buildings | 14 | 30 |
| 7.4 | Fire extinguishing and rescue operations | 6 | 11 |
| 8 | Safety during use | 17 | 33 |
| 9 | Sanitary and epidemiological requirements | 35 | 77 |
| 10 | Durability and maintainability | 7 | 8 |
| 11 | Energy saving | 6 | 14 |
| | Total | 151 | 356 |

The provisions of the standard were passed through an algorithm to form appropriate machine-readable rules based on them. As a result, 356 statements were received, which were divided into five groups (Figure 7).

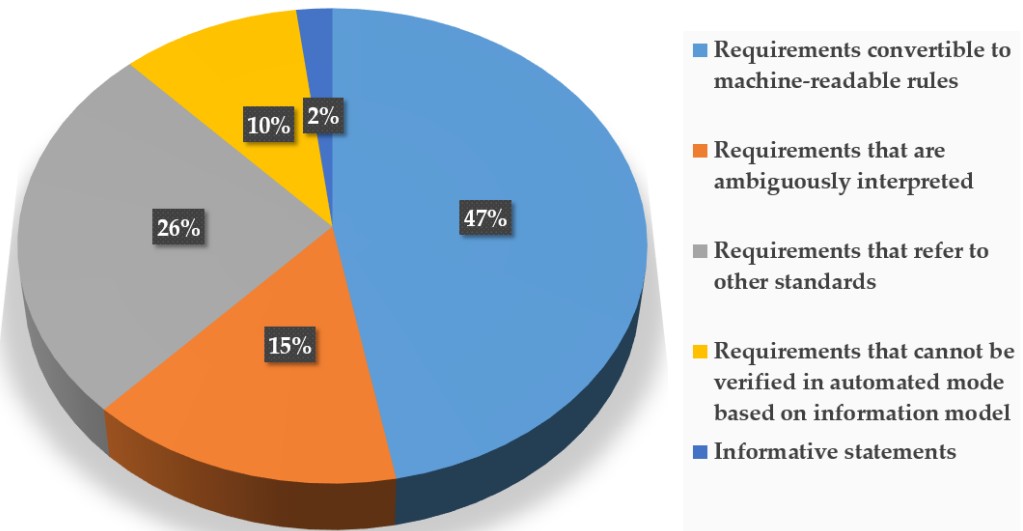

**Figure 7.** Types of statements of the standard SP 54.13330.2016.

In the following section, the main groups of statements were analyzed, as well as features inherent to this code, which limit machine-readable representation.

3.3.1. Requirements Suitable for Translation into Machine-Readable Format (47%)

A majority of statements of SP 54.13330.2016 are suitable for translation into a machine-readable format. This group of requirements can be considered in the context of three criteria (Table 2).

**Table 2.** Criteria applied to requirements suitable for translation into a machine-readable format.

| Criterion | The Essence of the Criterion | Algorithm of Translation |
|:---:|:---:|:---:|
| K1 | Rule ML formalization ability | Stage A1 feasibility |
| K2 | IFC translation ability | Stage A2 feasibility |
| K3 | No additional algorithms needed | - |

Requirements Corresponding to Criteria K1, K2, and K3

Section 7, provision 7.4.6: "In residential buildings (if divided into sections—in each section) with a height of more than 50 m, one of the elevators must provide transportation for fire departments".

This provision corresponds to one statement. Figures 8 and 9 show how this requirement was transformed after applying the algorithm. It is worth noting that to ensure the compactness of the presentation, the upper-level tags <Assert>-</Assert> and <Implies>-</Implies> were omitted. The resulting rule is ready for use for the automated verification of the building's information model.

```
                     Requirement after stage A1
<If>
        <And>
                <Atom>
                        <Var> building </Var>
                        <Rel> = </Rel>
                        <Ind> residential </Ind>
                </Atom>
                <Atom>
                        <Var> height </Var>
                        <Rel> greater than </Rel>
                        <Ind> 50 </Ind>
                </Atom>
        </And>
</If>
<Then>
        < Exists>
                <Atom>
                        <Var> elevator </Var>
                        <Rel> provides </Rel>
                        <Ind> transportation of fire departments </Ind>
                </Atom>
        </Exists>
</Then>
```

**Figure 8.** A machine-readable rule for provision 7.4.6 (after stage A1).

```
                     Requirement after stage A2
<If>
        <And>
                <Atom>
                        <Var> building </Var>
                        <Rel> = </Rel>
                        <Ind> residential </Ind>
                </Atom>
                <Atom>
                        <Var> height </Var>
                        <Rel> greater than </Rel>
                        <Ind> 50 </Ind>
                </Atom>
        </And>
</If>
<Then>
        < Exists>
                <Atom>
                        <Var> elevator </Var>
                        <Rel> provides </Rel>
                        <Ind> transportation of fire departments </Ind>
                </Atom>
        </Exists>
</Then>
```

**Figure 9.** A machine-readable rule for provision 7.4.6 (after stage A2).

Requirements Corresponding to Criteria K1 and K3

The standard also contains requirements for which only the first stage of the algorithm is feasible. It means that it is impossible to switch to IFC terminology, as there are no

corresponding IFC entities or properties in the standard scheme. In this study, the IFC version 4.2 scheme was considered [26].

Section 5, provision 5.8: "The height (from floor to ceiling) of living rooms and kitchens (kitchens and dining rooms) in climatic regions I-A, I-B, I-C, I-D, and II-A, determined by SP 131.13330, must be at least 2.7 m, and in other climatic regions, at least 2.5 m".

This provision contains two statements: the first refers to the regions I-A, I-B, I-G, I-D, and II-A and the second refers to all the others. If the first statement is considered, the reference to the SP 131.13330 standard is not essential here since the transition to this standard will only clarify the method of climatic region division. In our case, the climatic region is predetermined. Figure 10 shows how the utterance was transformed after applying the first stage of the algorithm.

```
                         Requirement after stage A1
<if>
        <and>
                <or>
                        <Atom>
                                <Var> climatic region </Var>
                                <Rel> equal </Rel>
                                <Ind> IA </ Ind >
                        </Atom>
                        <Atom>
                                <Var> climatic region </Var>
                                <Rel> equal </Rel>
                                <Ind> I -B </ Ind >
                        </Atom>
                        <Atom>
                                <Var> climatic region </Var>
                                <Rel> equal </Rel>
                                <Ind> I-C </ Ind >
                        </Atom>
                        <Atom>
                                <Var> climatic region </Var>
                                <Rel> equal </Rel>
                                <Ind> I-D </ Ind >
                        </Atom>
                        <Atom>
                                <Var> climatic subdicstrict </Var>
                                <Rel> equal </Rel>
                                <Ind> II-A </ Ind >
                        </Atom>
                </or>
                <or>
                        <Atom>
                                <Var> premise </Var>
                                <Rel> is </Rel>
                                <Ind> liviong </ Ind >
                        </Atom>
                        <Atom>
                                <Var> premise </Var>
                                <Rel> is </Rel>
                                <Ind> kitchen </ Ind >
                        </Atom>
                </or>
        </and>
</if>
<then>
        <Atom>
                <Var> ceiling heght of premise </Var>
                <Rel> is no less than </Rel>
                <Ind> 2,7 </ Ind >
        </Atom>
</then>
```

**Figure 10.** A machine-readable rule for provision 5.8 (after stage A1).

This requirement has been successfully formalized and translated into RuleML. However, the IFC scheme does not support the attribute "climatic region", which means it is not possible to translate the rule into IFC terminology or check it. In this case, additional user attributes for information model elements are required [36]. Nevertheless, the uniformity approach must be followed to avoid the application of different attributes for the same requirements.

Requirements Corresponding to Criteria K1 and K2

Section 7, provision 7.1: "Intersectional, inter-apartment walls, and partitions, as well as walls and partitions separating non-apartment corridors, halls, and lobbies from other premises, must comply with the requirements set out in Table 7.2." (see Table 3).

**Table 3.** Table 7.2 of the Construction code SP 54.13330.2016 [56].

| Enclosing Structures | The Minimum Fire Resistance Limit and the Permissible Fire Hazard Class for the Structure of the Building | | |
|---|---|---|---|
| | I-III, C0 и C1 | IV, C0 и C1 | IV, C2 |
| Intersectional wall | REI ** 45, K0 * | REI ** 15, K0 * | REI ** 15, K2 |
| Intersectional partition | EI 45, K0 * | EI 15, K0 * | EI 15, K2 |
| Inter-apartment wall | REI ** 30, K0 * | REI ** 15, K0 * | REI ** 15, K2 |
| Inter-apartment partition | EI 30, K0 * | EI 15, K0 * | EI 15, K2 |
| Walls separating non-apartment corridors from other premises | REI ** 45, K0 * | REI ** 15, K0 * | REI ** 15, K2 |
| Partitions separating non-apartment corridors, halls, and lobbies from other premises | EI 45, K0 * | EI 15, K0 * | EI 15, K2 |

* For C1-class buildings, K1 is allowed. ** For curtain walls, the fire resistance limit according to the limit state "loss of bearing capacity (R)" is not set.

This requirement has also been formalized and translated into RuleML (the translation is not presented in the article due to its large volume). Meanwhile, additional program code is required to make a preliminary search for these elements within the information model in order to determine the "intersectional walls", "inter-apartment walls", and other structures shown in Table 3.

3.3.2. Requirements Containing References to Other Documents, including Non-Regulatory Ones (26%)

Section 6, provision 6.1: "The foundations and supporting structures of an apartment building must be designed according to GOST 27751, SP 16.13330, SP 20.13330, SP 63.13330, and SP 70.13330".

In this provision, specific requirements for foundations and supporting structures are not established, but a list of codes in which they are presented is provided.

Section 5, provision 5.2: "In multi-apartment buildings of private housing areas according to [4,15] and commercial housing areas, the number of rooms and the area of apartments should be set in the design assignment, taking into account the specified minimum areas of apartments and the number of rooms".

This provision does not contain requirements for the areas and the number of apartments in multi-apartment buildings in private housing areas but indicates that these parameters should be set in the design assignment. References to the design assignment in code SP 54.13330.2016 are quite common.

Section 4 "General terms", provision 4.5: "In residential buildings, it is necessary to provide household drinking and hot water supply, sewerage, and drains in accordance with SP 30.13330 and SP 31.13330, and heating, ventilation, smoke protection in accordance with SP 60.13330".

This provision seems more complicated. On the one hand, it requires certain engineering infrastructure in the building, which can be translated into a rule and checked. On the other hand, there are no specific requirements for this infrastructure; only a link to the

standard containing them is indicated. In such cases, split the statement into two or more sub-statements and their individual check can be applied.

For verification of such types of requirements, two solutions are possible:

- Pre-replacement of the links with specific requirements from the mentioned documents and standards.
- Interconnection of the rule bases, which was formed by various documents, including regulatory ones. Thus, when the verification system switches to rules containing a link, it will automatically switch to the rule base with the referenced document.

Moreover, specific sections and provisions of standards to be verified are usually not specified, and that might be another problem. Thus, experts must first perform their searches in the mentioned documents.

### 3.3.3. Requirements with Uncertain Interpretation (15%)

Section 4, provision 4.3: "When designing and constructing a residential building, vital activity conditions of low-mobility groups of the people should be provided. These include the accessibility of the site, buildings and apartments constructed for the disabled and elderly people who use wheelchairs and disabled people with complete loss of vision and (or) hearing. The apartments for families with disabilities in this residential building should be set in the design assignment".

This provision establishes the need to comply with "conditions for the vital activity of low-mobility groups of the population", but these conditions are not specified.

Section 4, provision 4.8: "The width of the doors in the cabin of one of the elevators should ensure the passage of a wheelchair".

This provision does not provide a specific width for the doors or wheelchair, which makes the provision in its current form unsuitable for forming a verification rule.

The involvement of experts to clarify the relevant conditions is necessary to translate such requirements into a machine-readable format. The specification of these conditions, in this case, refers to SP 59.13330.2016 "Accessibility of buildings and structures for low-mobility groups of the population", which also implies an implicit reference to another standard.

### 3.3.4. Requirements That Cannot Be Verified Using the Information Model Basis (10%)

Section 8, provision 8.6: "Engineering equipment and instruments in case of possible seismic impacts must be securely fixed".

This provision contains a requirement, but this requirement can only be verified in situ and not by means of the information model. The same may be observed for the majority of statements attributed to this group.

### 3.3.5. Informative Statements (2%)

Informative statements do not contain requirements but may provide explanations for other statements, so their content is not subject to translation into machine-readable rules [56].

Section 4 "General terms", provision 4.1: "The rules for determination of the area of buildings and their premises, construction area, number of floors, and building volume in the design are given in Appendix A".

Explanation of the table in Appendix B of SP 54.13330.2016: "The table is based on the following calculation: there is 18 sq.m of the total area of the apartment per person, the height of the floor is 2.8 m, and the interval of movement of the elevators is 81–100 s".

### 4. Discussion and Conclusions

The distribution of different types of statements by sections of the standard is shown in Figure 11. Based on the statistics obtained, the following conclusions can be provided:

1. In four of the eight sections, the amount of requirements suitable for translation into machine-readable format is close to 50% or higher, with a maximum of 63% in the

"Fire safety" section. Analysis showed that attempts to verify fire safety requirements based on information models were the most frequent compared to others.

2.  In four of the eight sections, the number of requirements that cannot be verified based on the information model was less than 10%. The maximum number of such requirements (75%) is in the "Durability and maintainability" section, which is quite logical since the provisions of this section mostly relate to already-erected objects.

3.  Requirements containing references, requirements with uncertain interpretation, and informative requirements were encountered in all sections with approximately the same intensity.

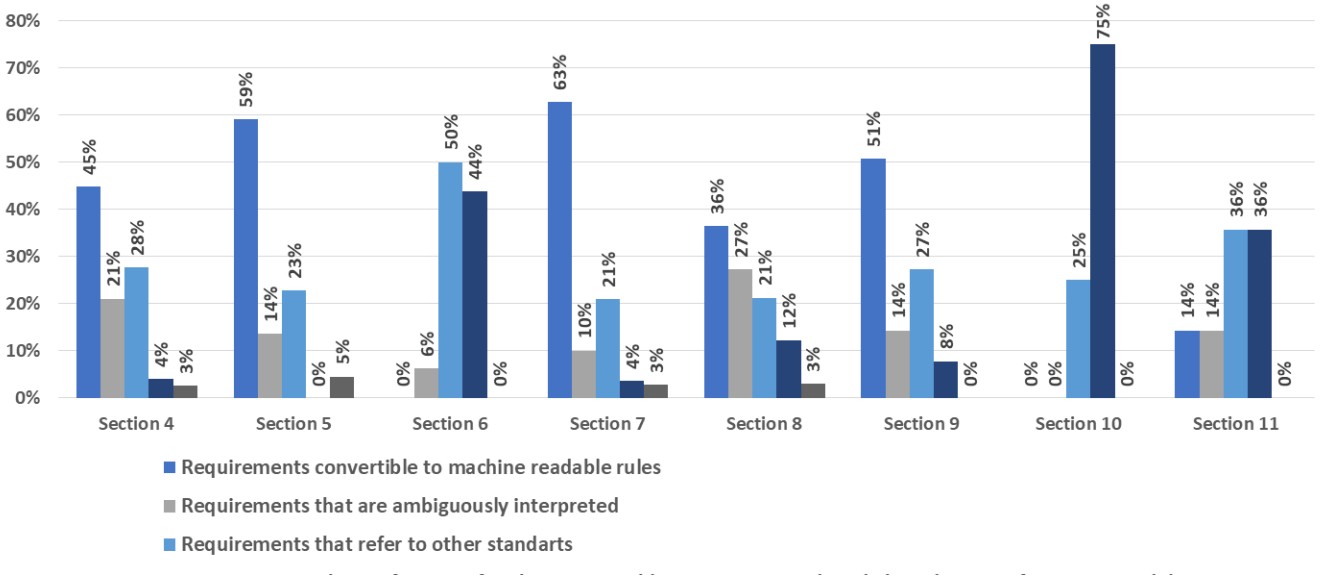

**Figure 11.** Distribution of statements in the sections of SP 54.13330.2016.

*4.1. Conclusions and Discussion Regarding the Suitability of the Developed Algorithm for Translation into a Machine-Readable Format for Normative Documents*

The algorithm for forming rules for checking information models of construction objects based on the rules of modeling language has fully justified itself.

1.  In examples, it is demonstrated that RuleML allows one to obtain a machine-readable representation of the requirements that is also understandable to humans.

2.  The analysis of RuleML syntax proved that it has the capability to establish a connection with the original document test. To do this, the @iri attribute can be used, the use of which will be considered in further studies.

3.  To use RuleML, both text editors and existing development tools were tested, partially automating this process. The authors developed their own environment in Excel using Visual Basic for Applications with a more user-friendly interface.

4.  The RuleML specification is indeed published on the official website, which allowed it to be studied and used for its own needs.

5.  The approach is indeed scalable, which is confirmed by its specification. However, when translating this normative document, the use of language extensions (such as MathML or Legal RuleML) was irrelevant.

6.  RuleML does not depend on a specific software or hardware platform, and the rules written on it can be processed for use in various verification programs, as well as for training neural networks, which will be tested in the future. The possibility of supporting rules and the interactions of one rule base with others also represents a direction for further research.

7. By using the appropriate tags of logical operations and quantifiers, RuleML rules can display an infinite number of requirements, including nested conditions and branching of alternative contexts within a specific area of knowledge.

*4.2. Conclusions and Discussion Regarding the Suitability of the Standard for Translation into Machine-Readable Format*

It is generally worth noting that a significant part of the requirements are suitable for translation into a machine-readable format, and the limiting factors could be overcome both by attracting experts and by programming tools. A conducted study showed that the algorithm for forming rules for verification of building information models based on RuleML language is a reliable tool.

However, despite the above, the translation process may still be quite time-consuming. Thus, the following recommendations may be given on how to re-arrange regulatory documents:

1. If a requirement within any regulatory document contains a link to another regulatory document or documents, the reference should specify the section number and the provision number of the cited document(s). It is not possible to abandon the links; otherwise, it would mean replacing the link with the entire text. In this case, alterations in the source of the requirement will require alterations in all documents where this requirement was stated instead of referred to.

2. Clear requirements instead of vague interpretations:
   - The object of verification (an element or its property), the required value, and the relationship linking the object and the value;
   - The first and the second objects of verification, and the relationship that is required between them.

3. The unity of terminology is crucial both within one standard and within the entire complex of codes for construction, predicting in advance their relationship with IFC terms.

It is worth noting that the results and conclusions given in this article can be extended to other Russian standards. The individual problems identified above echo those identified by foreign colleagues [37], although their list and description are more widely presented in this article, which also indicates the applicability of the research results to foreign standards.

**Funding:** This work was financially supported by the Ministry of Science and Higher Education (grant No. 075-15-2021-686). All tests were carried out using research equipment at The Head Regional Shared Research Facilities of the Moscow State University of Civil Engineering.

**Data Availability Statement:** Not applicable.

**Conflicts of Interest:** The author declares no conflict of interest.

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
