# Peer review of "Features of Regulation Document Translation into a Machine-Readable Format within the Verification of Building Information Models"

_2673-4109, doi:10.3390/civileng4020022_

Round 1
Reviewer 1 Report
Overall the paper is interesting - but I think several changes are needed to make it into a publishable scientific paper:
-The authors state that they propose an algorithm - I did not see any evidence of this automation or how it works documented
-Please describe the differences between your work in LegalML and previous work using similar techniques i.e. RASE, N3 logic etc...
-Please reframe the paper using a scientific methodology - set out an aim/research question and then use your automated approach to test against it i.e. accuracy of output, etc....
Author Response
Dear reviewer,
I would like to thank you for your valuable comments, which helped to improve the quality of the article. Please find a detailed description of the comments and their consideration in the article below. All changes are marked in yellow in the revised version of the article.
Comment 1: The authors state that they propose an algorithm - I did not see any evidence of this automation or how it works documented
Reply to the comment 1: Subsection "3.2 Automation of rule formation algorithm for RuleML-based verification of building information models" was added to the section "3. Results".
Comment 2: Please describe the differences between your work in LegalML and previous work using similar techniques i.e. RASE, N3 logic etc...
Reply to comment 2: The appropriate paragraph was added to the section 2.1.
Comment 3: Please reframe the paper using a scientific methodology - set out an aim/research question and then use your automated approach to test against it i.e. accuracy of output, etc....
Reply to comment 3: The "Introduction" section was extended, which gives more details of what will be revealed in the article. At the end of the "Introduction" section, the purpose was formulated in more detail, and the tasks corresponding to the sections of the article were set.
Thank you for your consideration of this manuscript.
Reviewer 2 Report
In this paper a proposal about the translation of provisions presented in regulatory documents into a machine-readable format is developed. For this, some algorithms are proposed, written on the RuleML rule modeling language.
I consider that this paper is well-written, and the use of this language seems adequate in order to achieve the desired effects. In my opinion I do not have suggestions. The author divides into several statement groups the information that has to be analyzed, and describe the complexity of the problem for each case.
This paper could be interesting for readers, but I miss more information about the quality control of the procedure.
Author Response
Dear reviewer,
I would like to thank you for your valuable comments, which helped to improve the quality of the article. Please find a detailed description of the comments and their consideration in the article below. All changes are marked in yellow in the revised version of the article.
Comment 1: This paper could be interesting for readers, but I miss more information about the quality control of the procedure.
Reply to the comment 1: The quality control of the procedure will be performed using Inter-rater reliability (IRR) as a subject of further research.
Thank you for your consideration of this manuscript.
Reviewer 3 Report
This is an interesting paper on a topical subject. I am not an expert in this area, but I’ve co-supervised a PhD (1) on the topic. I therefore feel reasonably comfortable about reviewing this paper. It is generally understandable but I often did not follow what being said. It may be that the author expects readers to be as familiar with the topic as she is. This is certainly not the case for myself.
Many of my concerns would probably be allied is the paper were to be proof-read by an native English speaker conversant with BIM-enabled code checking. The author uses long sentences further impede understanding.
The manner in which the paper is organized could be improved. Very little ‘signposting’ is provided to help readers navigate the text. It was only when I got to p.4 that I read what the purpose of the paper was. At this point I expected to read how the paper was structured but no explanation was given.
I have inserted several post-its to the text I’ve highlighted in the pdf. Please see the attachment.
My main concern relates to methodology. I found section 1 interesting and informative. It provides adequate background - but readers are left to ‘join the dots’ themselves’. I could not locate a clear research question that the author sought to address. Some research aims are stated in the abstract… but exactly what is this study seeking to establish?
My concerns extend to the methods section. I acknowledge that there are word / page limits that constrain what can be said… but the justification for the approach adopted could be strengthened. Yes, the author mentions her previous investigations (44) and (25) on lines 159 and 179. Is this sufficient? Perhaps this paper could better be described as a case study of the use of RuleML (?)
Also, and importantly, it appears that no third-party validation was conducted. Here I’m referring to the manual tagging / interpreting of rules that I think were conducted, presumably by the author. In such cases it’s usual for someone else to replicate the work. Inter-rater reliability (IRR) is usual in qualitative studies and surely this applies here as well? No details of this are provided.
Finally, I am not competent to comment on the detailed explanations provided in section 3. I will defer to the recommendations of others in this regard.
On a minor (and admittedly petty) note, I have an aversion to the use of ‘we’ and ‘you’. This is a sole authored paper, so the author cannot be referring to colleagues. It is presumptuous to anticipate what readers feel / agree with.
(1) Shih, S. Y. (2015). Challenges associated with implementing BIM-enabled code-checking systems within the design process. PhD thesis submitted to The University of Newcastle, Australia

Author Response
Dear reviewer,
I would like to thank you for your valuable comments, which helped to improve the quality of the article. Please find a detailed description of the comments and their consideration in the article below. All changes are marked in yellow in the revised version of the article. I also attached first version manuscript with your post-its and my comments to them
Comment 1: Many of my concerns would probably be allied is the paper were to be proof-read by an native English speaker conversant with BIM-enabled code checking. The author uses long sentences further impede understanding.
The manner in which the paper is organized could be improved. Very little ‘signposting’ is provided to help readers navigate the text. It was only when I got to p.4 that I read what the purpose of the paper was. At this point I expected to read how the paper was structured but no explanation was given.
Reply to the comment 1: The "Introduction" section was extended, which gives more details of what will be revealed in the article. At the end of the "Introduction" section, the purpose was formulated in more detail, and the tasks corresponding to the sections of the article were set.
Comment 2: I have inserted several post-its to the text I’ve highlighted in the pdf. Please see the attachment.
Reply to the comment 2: The manuscript was carefully revised according to the post-its
Comment 3; My main concern relates to methodology. I found section 1 interesting and informative. It provides adequate background - but readers are left to ‘join the dots’ themselves’. I could not locate a clear research question that the author sought to address. Some research aims are stated in the abstract… but exactly what is this study seeking to establish?
Reply to the comment 3: At the end of the "Introduction" section, the purpose was formulated in more detail, and the tasks corresponding to the sections of the article were set
Comment 4: My concerns extend to the methods section. I acknowledge that there are word / page limits that constrain what can be said… but the justification for the approach adopted could be strengthened. Yes, the author mentions her previous investigations (44) and (25) on lines 159 and 179. Is this sufficient? Perhaps this paper could better be described as a case study of the use of RuleML (?)
Reply to the comment 4: The principles of Rule ML used in the study are focused in more detail in the "Results" section within the description of the rule formation algorithm for RuleML-based verification of building information models.
Comment 5: Also, and importantly, it appears that no third-party validation was conducted. Here I’m referring to the manual tagging / interpreting of rules that I think were conducted, presumably by the author. In such cases it’s usual for someone else to replicate the work. Inter-rater reliability (IRR) is usual in qualitative studies and surely this applies here as well? No details of this are provided.
Reply to the comment 5: Thanks for the recommendation. Since the markup of the standard is a labor-intensive and time-consuming process, verification using Inter-rater reliability (IRR) will be the subject of further research.
Comment 6: On a minor (and admittedly petty) note, I have an aversion to the use of ‘we’ and ‘you’. This is a sole authored paper, so the author cannot be referring to colleagues. It is presumptuous to anticipate what readers feel / agree with.
Reply to the comment 6: The language of the paper has been corrected.
Thank you for your consideration of this manuscript.

Round 2
Reviewer 1 Report
Thank you to the authors the paper is now much improved. I have only one remaining comments. Early on in the paper the authors describe one of their keys steps as
"Forming conclusions regarding the suitability of the developed algorithm for translation into a machine-readable format of regulatory documents."
I would really like to see it laid out how they will do this and how they will analyse if it is suitable. What metrics will they set and compare against?
Author Response
Dear reviewer,
thank you very much for your valuable comments. According to them, the criteria of RuleML application and conclusions according to this criteria are marked in green in the text.
The language of the manuscript has also carefully been revised
Reviewer 3 Report
My suggestions have been responded to. I have nothing further to add
Author Response
Dear reviewer,
Thank you very much for your valuable recommendations for improving the manuscript